# Investigation of Pre-Aged Hardening Single-Point Incremental Forming Process and Mechanical Properties of AA6061 Aluminum Alloy

**DOI:** 10.3390/ma16114154

**Published:** 2023-06-02

**Authors:** Yao Zhang, Zhichao Zhang, Yan Li, Lan Hu, Qiu Pang, Zhili Hu

**Affiliations:** 1Hubei Longzhong Laboratory, Wuhan University of Technology, Xiangyang 441000, China; 2Shanghai Aerospace Equipments Manufacturer Co., Ltd., Shanghai 200245, China; 3Department of Mechanical and Electrical Engineering, Wuhan Donghu University, Wuhan 430212, China; 4Hubei Key Laboratory of Advanced Technology for Automotive Components, Wuhan University of Technology, Wuhan 430070, China; 5Hubei Engineering Research Center for Green & Precision Material Forming, Wuhan University of Technology, Wuhan 430070, China

**Keywords:** aluminum alloy, PH-SPIF, forming limits, mechanical properties, microstructure

## Abstract

Currently, the single-point incremental forming process often faces issues such as insufficient formability of the sheet metal and low strength of the formed parts. To address this problem, this study proposes a pre-aged hardening single-point incremental forming (PH-SPIF) process that offers several notable benefits, including shortened procedures, reduced energy consumption, and increased sheet forming limits while maintaining high mechanical properties and geometric accuracy in formed components. To investigate forming limits, an Al-Mg-Si alloy was used to form different wall angles during the PH-SPIF process. Differential scanning calorimetry (DSC) and transmission electron microscopy (TEM) analyses were conducted to characterize microstructure evolution during the PH-SPIF process. The results demonstrate that the PH-SPIF process can achieve a forming limit angle of up to 62°, with excellent geometric accuracy, and hardened component hardness reaching up to 128.5 HV, surpassing the strength of the AA6061-T6 alloy. The DSC and TEM analyses reveal numerous pre-existing thermostable GP zones in the pre-aged hardening alloys, which undergo transformation into dispersed β” phases during the forming procedure, leading to the entanglement of numerous dislocations. The dual effects of phase transformation and plastic deformation during the PH-SPIF process significantly contribute to the desirable mechanical properties of the formed components.

## 1. Introduction

Single-point increment forming (SPIF) is a flexible plastic forming technique that employs a tool to follow a predefined trajectory and form the sheet layer by layer [1], similar to 3D printing. The process does not require the use of specialized dies or only needs simple dies to support [2], enabling different parts to be formed by modifying the program code that controls tool movement. Thus, the procedure significantly reduces the expense and time consumed in creating molds, minimizing the cost of production. Furthermore, SPIF is well-suited for the manufacturing of one-off or small-batch products and for developing new merchandise.

Compared to conventional stamping methods, single-point incremental forming (SPIF) often provides superior formability due to the local forces exerted on the sheet [3], and its forming limit is typically determined by the forming wall angle as the wall thickness satisfies the cosine theorem during the forming process. When the wall angle of the part to be formed exceeds the forming limit of the sheet, the sheet may need to be machined gradually to the desired shape using the multi-pass single-point incremental forming process (MSPIF). However, the forming time required for MSPIF is prolonged, and it is difficult to determine the forming times and shape of the forming, making the process challenging. Therefore, enhancing the sheet-forming limit has become a topic of interest among scholars both domestically and internationally.

Hussain et al. [4] investigated the effect of SPIF parameters such as tool diameter, vertical pitch, and forming speed on the formability and surface roughness of annealed and pre-aged AA2024 aluminum sheets. They found that vertical pitch and tool diameter had a more significant impact on formability, while forming speed had different effects on the sheet in different states. Since higher forming temperatures can effectively increase the formability of aluminum alloys, many researchers have utilized auxiliary heating methods, such as hot air convection [5], laser heating [6,7,8], and electric conduction heating [9,10,11], during the forming process to enhance the forming limit of the sheet. When using laser heating, the high-power density of the laser may result in localized stress concentration within the alloy, leading to the generation of cracks and subsequent formation of defects [12]. Increasing the tool speed during forming can also effectively raise the forming temperature [13]. Otsu et al. [14] investigated the formability of AA5052-H34 aluminum alloy sheets using tool speeds ranging from 2000–10,000 rpm and discovered that speeds above 7000 rpm significantly improved the formability of the material through temperature rise and dynamic recrystallization. Buffa et al. [15] formed AA1050-O, AA1050-H24, and AA6082-T6 aluminum alloy sheets at different rotational speeds and plotted the sheet-forming limit curves at various rotational speeds.

Nevertheless, Ambrogio et al. [10] demonstrated that the use of local heating techniques resulted in an increase in wall angle but a decrease in surface roughness in SPIF. Temperature fluctuations also affect the microstructure and properties of the sheet, which have an impact on the mechanical properties of the resulting pieces. Gou et al. [16] also indicated that the tensile strength of the weld joint could reach 420 MPa at a lower heat, and the service life is improved accordingly. Reasonably controlling the microstructure of aluminum alloy and reducing stress concentration can effectively enhance the fracture toughness of aluminum alloy components, consequently improving their formability [17]. Thus, many researchers have started to focus on studying the laws governing the evolution of the microstructure during the forming process when considering forming limits. Moises et al. [18] proposed a neural network-based approach for aluminum alloy material design, which can significantly improve the formability of sheet metal during the single-point incremental forming process. Barnwal et al. [19] conducted a comprehensive study on the deformation of AA6061 aluminum alloy plates and discovered that plastic anisotropy significantly affects the microstructure and texture development in various directions of AA-6061 alloy plates during SPIF, where metal flow is always perpendicular to the direction of tool movement. Chen et al. [20] found that dynamically recrystallized parts produced equiaxed grains with a non-uniform microstructure along the thickness direction during SPIF of the AA2024-T3 sheet. To address this issue, they proposed a two-stage single-point incremental forming method that successfully obtained uniform grains in the thickness direction of the sheet, inducing homogeneous fine precipitates, pinning effects, and more grain boundaries within the grains, leading to an increase in the strength of the formed part. Chen et al. [21] improved the mechanical properties of joints by annealing at 650 °C and 850 °C and significantly increased joint strength using precipitation strengthening. Xu et al. [22] proposed an effective annealing treatment process by studying the structural evolution and mechanical properties of three annealed state alloys through annealing heat treatment. Mohammadi et al. [23] found that annealed conditions increased the maximum forming angle of the AA2024 aluminum sheet compared to the aging treatment. However, after the annealing treatment, the hardness of the sheet was greatly reduced, necessitating a subsequent heat treatment to recover it. Nevertheless, the subsequent heat treatment may cause the formed part to distort, resulting in poor geometric accuracy. Ghaferi et al. [24] compared the forming limits of the AA6061 sheet under different heat treatment cycles and found that proper heat treatment could enhance the formability and mechanical properties of the sheet. Khan et al. [25] emphasized that the sequence of the forming process and the state of heat treatment have a significant impact on the accuracy, formability, and residual stresses in the SPIF of the AA2219 aluminum sheet.

The primary process for forming high-strength aluminum alloy components traditionally involves shaping the part and subsequently applying heat treatment to enhance its microstructural properties. However, this method is associated with several issues, including heat treatment deformation, loss of accuracy, prolonged procedures, and low efficiency, which may ultimately render aluminum alloy impractical for industrial production. The recent proposal by Hua [26,27] of a pre-aged hardening warm forming process for the AA7075 aluminum alloy sheet has shown promising results, wherein the material is solution heat-treated and quenched to obtain W-temper aluminum alloy sheet, which is then immediately pre-reinforced to obtain Tx sheet. The resulting material exhibits excellent mechanical properties and strengths exceeding those of AA7075-T6, without requiring any subsequent heat treatment. Building upon this research, the current study proposes a pre-aged hardening single-point incremental forming process (PH-SPIF) utilizing AA6061 aluminum alloy. A brief diagram of the process chain is shown in Figure 1. The investigation focuses on determining the single-point incremental forming limit, plastic flow behaviors, mechanical properties, and microstructure–strength relationships of this material. To further analyze the microstructure of the formed samples, transmission electron microscopy (TEM) and differential scanning calorimetry (DSC) techniques are employed.

## 2. Experiment

### 2.1. Material and Specimen Design

The Al-Mg-Si alloy used in this study was obtained from Alnan Aluminum Co., Ltd. (Nanning, China) in the peak-aged T6 temper with a sheet thickness of 2 mm. In accordance with the Chinese National Standard, Table 1 displays the chemical composition of the alloy, whereas Table 2 delineates its physical properties. Tensile test specimens were designed according to the schematic shown in Figure 2 and fabricated using wire-electrode cutting. Rectangular samples measuring 8 mm × 8 mm were utilized for TEM microscopy, undergoing sandpaper thinning to achieve a thickness of 80 nm before being subjected to ion-beam thinning. Samples for hardness testing were ground and polished to dimensions of 20 mm × 8 mm prior to the hardness tests. DSC tests utilized small disks with a diameter of 1.2 mm and weighing between 10–15 mg.

### 2.2. Tensile Test

The tensile samples underwent a heat treatment process, including a 535 °C treatment for 40 min followed by an immediate quenching in cold water. Subsequently, the samples were promptly transferred to a 101-113S electric blast drying oven produced by Shanghai Lichenbangxi Instrument Technology Co., Ltd. (Shanghai, China), for subsequent holding procedures. The holding temperatures used in this study were 130 °C, 140 °C, and 150 °C, while the corresponding holding times were either 12 or 18 h. A schematic of the plate material treatment process is presented in Figure 3. Each sample group was subjected to three tensile tests to ensure the reproducibility of results, and stress–strain curves were generated to analyze their mechanical properties.

### 2.3. Forming Process

The sheet metal utilized in this study had dimensions of 270 mm × 270 mm × 2 mm and was subjected to the same heat treatment process as the 150 °C-12 h tensile sample. Single-point incremental forming experiments were conducted on a CNC milling machine, which was equipped with a SINUMERIK 808D CNC control system. Special fixtures were utilized to constrain all four edges of the sheet with all degrees of freedom, as depicted in Figure 4. The tool used in this study had a radius of 5 mm and was fabricated using H13 steel. During the forming process, temperature measurements were taken using a handheld thermal imaging camera, which captured the temperature at the contact point between the tool and the sheet material, and plotted the time-temperature curve. The tool trajectory was selected as a spiral trajectory, with the motion coordinates being derived by the mathematical software Matlab. The CNC code was generated from the resultant tool path and transferred to the CNC machine. The forming area was defined as a radius 80 mm circle on the upper surface of the sheet. To reduce testing times, a variable angle cone trajectory with the generatrix as an arc was chosen for forming, with the center of the sheet serving as the origin and the equation of the arc trajectory defined as ${\left(x-120\right)}^{2}+{\left(y+80\right)}^{2}=8000$. The forming depth was set at 60 mm, with the forming angle growing from 26.6° to 77.1°, as demonstrated in Figure 5. Truncated cone trajectories with different wall angles and a 40 mm forming depth were employed to verify the conclusions. The process parameters utilized are listed in Table 3, where the PA state refers to the state of the sheet after pre-aged hardening treatment, and the T6 state refers to the peak-aged state. The two aforementioned alloy states exhibit discernible dissimilarity solely in the configuration of their forming wall angles, while all other pertinent process parameters remain entirely congruent. Furthermore, the formed part was scanned using a handheld 3D laser scanner, specifically the CREAFORM HandySCAN 700, and subsequently compared with the CAD model to evaluate the geometric accuracy of the forming process.

### 2.4. Hardness Testing and Microstructural Observations

The plate was cut in the radial direction after forming. To prevent any further aging of the samples during the mounting process, cold mounting was employed. Standard grinding and mechanical polishing methods were used to prepare specimens for hardness testing. Vickers hardness testing was performed using an HV-1000A indentation machine with an indentation load of 200 g and a dwell time of 10 s. Each measurement point was tested five times, and the average value was calculated, resulting in a total of fifteen mean values. Images were plotted to visualize the results. For comparison purposes, fifteen points were tested for each of the 6061-T6 and 6061-PA, and the average was calculated. To investigate the microstructure and types of precipitates present, differential scanning calorimetry (DSC) analysis and transmission electron microscope (TEM) observations were conducted. DSC and TEM samples were taken from the formed area of the part. The DSC tests were conducted on a Netzsch DSC 214 calorimeter at a heating rate of 10 °C/min. TEM observations were made incident from the <001> direction, and dark field imaging was used to observe dislocations and precipitations. High-resolution transmission electron microscopy was performed subsequently to measure the precipitation dimension parameters and obtain their diffraction pattern by Fourier transform.

## 3. Results

### 3.1. Tensile Properties of PA-State Sheet at Different Parameters

In order to study the mechanical properties of pre-aged hardening sheets, uniaxial tensile tests were performed on sheets subjected to various treatments, including holding at 130 °C-12 h, 130 °C-18 h, 140 °C-12 h, 140 °C-18 h, 150 °C-12 h, and 150 °C-18 h. The stress–strain curves and elongation bar diagrams are presented in Figure 6a,b, respectively. The standard deviation of tensile strength for each test group does not exceed 7.14 MPa. “O-temper” aluminum alloy refers to an aluminum alloy that has been fully annealed through a process of heat treatment. The results show that as compared to holding at the same temperature for 12 h, the hardening effect was more significant with an 18 h hold time. Moreover, an overall upward trend was observed in the yield strength of the plate material as the holding temperature increased from 130 °C to 150 °C, increasing from 211 MPa to 271 MPa. However, this increase in yield strength was accompanied by a decrease in elongation to varying degrees. The two groups of sheets held at 150 °C exhibited higher mechanical properties, with yield strengths of 271 MPa and 279 MPa, respectively. However, there was a significant difference in elongation between these groups. Therefore, the optimal strengthening parameter can be considered to be 150 °C for 12 h.

### 3.2. Formed Parts with Different Wall Angles

By utilizing a variable angle cone trajectory, the PA-state sheet was formed, which resulted in sheet failure at a forming depth of 39.804 mm. To determine the theoretical forming angle, the trajectory tangent was calculated, resulting in an angle of 63.3°. Next, forming angles of 63° and 62° were selected, with the sheet rupturing at a forming angle of 63° but being formed successfully at a forming angle of 62°, leading to the ultimate forming angle of the sheet being 62° as shown in Figure 7. The results of the T6 state sheet forming are presented in Figure 8, wherein fracture occurs with a 55° wall angle while the sheet is formed smoothly under a forming angle of 53°. The forming limit angles for different sheets are illustrated in Figure 9. The PA state sheet’s formation is consistent with that of 3003-O and 5754-O reported in the literature [28,29]. Figure 10 illustrates the cloud diagram comparison between the formed part and the CAD model evaluated under the tolerance of ±0.3 mm, which was generated using Geomagic Control X 64 software. It can be observed that the formed part and the side wall surface of the model fit completely, with 77.25% of the area within the tolerance range, meeting the requirement of ±1 mm; furthermore, the geometric accuracy of the formed part is high, with maximum upper and lower deviations of 0.69 mm and −0.81 mm, respectively.

### 3.3. Hardness of Forming Plate

Figure 11 depicts a comparison of the hardness of the component following forming by the PH-SPIF process with that of the T6 and PA states. The average hardness values corresponding to the PA and T6 states were measured to be 100.5 HV and 109.1 HV, respectively. The PHF state (post-forming part state) exhibits greater hardness, with its distribution aligning with the applied strain. Notably, the hardness tends to increase at higher strains and can attain values as high as 128.5 HV and 128.3 HV, with an average value of 120.9 HV. This represents a significant improvement over the T6 temper, exceeding it by 111%.

### 3.4. Microstructure of the Formed Sheet

Figure 12 shows the DSC test results for the W-temper alloy (solution heat-treated state) and manufactured PHF-temper sheet. The W-temper curve displays five peaks (A, B, C, D, and E), whereas the PHF-temper curve only has three apparent peaks (A, B, and C). A large number of dislocations were observed in the PHF state alloy through transmission electron microscopy, as shown in Figure 13. High-resolution TEM observations on both PA and PHF states of the alloy revealed that a significant amount of stable Guinier–Preston (GP) zones were present in the PA state alloy while strengthening phase β” was found in the PHF state alloy, as depicted in Figure 14.

## 4. Discussion

### 4.1. Formability

The SPIF process has been observed to offer superior forming limits compared to conventional press forming, owing to the significant shear deformation that occurs along the thickness direction of the sheet and the local plastic deformation resulting from the serrated strain path [30]. The formability of this process is strongly influenced by the tool radius: while smaller tool radii tend to concentrate the strain in the deformed area, larger radii distribute it over a wider region, thereby reducing formability and making the process more akin to conventional stamping [28]. However, a tool with a radius of 5 mm has been shown to effectively prevent necking at the beginning of forming, delay rupture, and maximize the formability of the alloy [31]. Additionally, a vertical pitch of 0.5 mm has been found to increase geometric accuracy, surface quality, and forming limit [32]. By decreasing positive friction at a spindle speed of 355 rpm, material formability can be enhanced [13], while the use of motor oil with superior flow properties can significantly reduce the friction between the tool and the sheet, leading to an improvement in forming limit [33]. Moreover, the friction generated during the forming process generates heat, which does not significantly affect grain size but promotes thermally activated dislocation movement, thereby increasing formability [34].

The microstructural characteristics of metallic materials play a vital role in determining their sheet-forming behavior. To quantify a metal’s ability to withstand uniform plastic deformation, engineers often use the strain hardening index (*n*). In the context of the AA6061 aluminum alloy sheet, the literature has extensively investigated the impact of *n* on its formability [35]. The findings indicate that an increase in the strain hardening index factor leads to a corresponding improvement in the sheet’s formability. Therefore, it can be concluded that enhancing the strain hardening index is an effective approach to enhance the sheet forming performance of AA-6061 aluminum alloy.
(1)$$\sigma =K\varepsilon^{n}$$
where σ and ε represent the true stress and true strain. n is the strain hardening index and K is the hardening coefficient. By taking the logarithm of the two sides of Equation (1), the following equation can be found:(2)lnσ=nlnε+lnK

By plotting the relationship curve of lnσ−lnε, the strain hardening index values for different states of the AA-6061 aluminum alloy sheet were determined. Specifically, the PA state had an *n* value of 0.15, which was greater than that of the T6 temper (0.05) and slightly less than that of the O temper (0.21). These results align with those obtained from uniaxial tensile testing.

Plastic deformation of a plate can cause microstructure instability, whereas elevated temperatures increase atomic diffusion capabilities and promote metal recovery. Figure 15 depicts the temperature curve obtained using a handheld thermal imaging camera during the forming process. The sheet metal was formed at room temperature, and with the accumulation of heat, the temperature gradually rose to 160 °C. The AA6xxx aluminum alloy exhibits excellent formability as its formability increases with rising temperatures below 200 °C [36,37]. In this case, the sheet metal will experience partial dynamic recrystallization and recovery phenomena [38]. The thickness of the sheet in the forming process follows the cosine theorem, which is widely accepted [39]. As the forming angle θ increases, the sheet progressively thins. If the forming angle exceeds the maximum forming angle $\theta_{max}$, the excessive thinning of the metal sheet can result in tensile stresses exceeding critical values, leading to sheet rupture. In the PA state, precipitation has a less obstructive effect on dislocations, resulting in more uniform stress distribution across the sheet. This leads to ultimate tensile stress being higher in the PA state than in the T6 temper. Therefore, the formability of the PA state is significantly greater than that of the T6 temper.

### 4.2. Mechanical Behavior before and after PH-SPIF

The size of the grain significantly influences a component’s mechanical characteristics. The shear stress between the forming tool and the sheet during the forming process affects grain refinement, and larger forming angles result in smaller grain sizes [40]. The gradually rising formation temperature does not affect grain size [35]. Plastic deformation is known to induce changes in microstructure, such as grain refinement and an increase in dislocation density [41]. As the tool rotates and feeds downwards, the sheet deforms, creating many dislocations. Around the precipitate phase, a significant stress field is formed due to the entangled dislocations, high-density dislocation wall, and dislocation ring, which restrict subsequent moving dislocations’ motion and cause a back stress strengthening effect. A large number of dislocations are observed to accumulate at grain boundaries, as depicted in Figure 13c. This considerable increase in dislocation resistance significantly limits dislocation glide and results in substantial work hardening.

Differential scanning calorimetry (DSC) can provide a reliable indicator of variations in an alloy’s precipitation sequence during the warming process. Precipitation strengthening is a highly complex nanoscale phenomenon that imparts strength to heat-treatable aluminum alloys such as 2xxx, 6xxx, and 7xxx [42]. The precipitation sequence for 6xxx aluminum alloys is widely accepted [43] as a supersaturated solid solution (SSSS) → GP zones → β” → β’ → β. The W-tempered alloy exhibits four distinct precipitation peaks and one dissolution peak between 50–550 °C. Peak “A” corresponds to the GP zone precipitation peak. The precipitation peak of the β” phase is peak “B”. Peaks “C” and “D” represent the precipitation peaks of the β’ and β’ phases, respectively, and peak “E” signifies the dissolution peak related to the dissolution of the β phase.

The formed PHF state alloy’s DSC curve exhibits two distinct exothermic peaks and a heat-absorbing peak at temperatures between 50 and 550 °C. The peaks “A” and “B”, which are relevant to the precipitation of β’ and β phases, respectively, occur at approximately 320 and 420 °C. The β phase then starts to dissolve, reaching a peak at around 490 °C. Unlike the W-temper alloy, the PHF state has a low temperature when precipitating the same second phase due to the accelerating effect of dislocations on precipitate formation [44,45]. Consequently, the dislocation-induced decrease in the precipitation temperature of β” avoids the detection of the PHF state alloy’s β” phase precipitation peak.

Precipitation strengthening represents the primary method for enhancing the strength of AA6061 aluminum alloys, and multiple investigations have demonstrated that precipitate structure and size play a dominant role in influencing the strength of Al-Mg-Si alloys [44]. During the aging treatment, vacancies result in solute atoms deflecting and eventually diffusing to precipitate in the second phase, as the supersaturated solid solution becomes unstable. Although dynamic recovery can partially counteract work hardening, the sheet’s mechanical properties are maintained due to the constant forming temperature of less than 200 °C [36].

Figure 14a illustrates the presence of numerous stable GP zones in the PA state of the sheet that can serve as direct nucleation cores for β” phase formation during the forming process [46]. This phenomenon results in numerous β” reinforced phases being present in the produced sheet, as shown in Figure 14b,c. The dual reinforcement effect of work hardening and phase transformation strengthening leads to the formed sheet being significantly harder than the PA sheet and even exceeding the T6 alloy.

## 5. Conclusions

In this study, AA6061 aluminum alloy was formed with different wall angles utilizing the pre-aged hardening single-point incremental forming process (PH-SPIF). The investigation of forming limits, post-forming mechanical properties, and phase transformation during the forming process led to the following conclusions:The proposed pre-aged hardening single-point incremental forming process (PH-SPIF) allows direct sheet forming after pre-aged hardening treatment. The results indicate that the forming limit angle of the AA6061 sheet under this process can reach 62°. The produced component has an average hardness of 120.9 HV, which exceeds the T6 state’s value of 109 HV and does not require additional heat treatment. The final tolerance of the manufactured component is 0.69 mm and −0.81 mm, satisfying the criterion of 1 mm. The PH-SPIF process exhibits high forming limits, high performance of the formed component, good mechanical properties, and geometric accuracy.Under the process parameters of 0.5 mm vertical pitch, 5 mm tool radius, 355 rpm rotational speed, and 1500 mm/min feed rate, the sheet temperature gradually increases to 160 °C, leading to a certain amount of dynamic recovery. Moreover, the excellent plasticity of the PA-state sheet and its point contact forming result in a notable increase in the forming limit.The dominant microstructural evolution of AA6061 aluminum alloy in the PH-SPIF process is the transformation of GP zones → β” phase. The microstructure of the pre-aged hardening alloy primarily consists of GP zones. During the forming process, a large number of GP zones transform into the sub-stable phase β”. The process characteristics lead to grain size refinement and the formation of high-density dislocation walls and rings within the sheet. Fine-grain strengthening, work-hardening, and transformation strengthening contribute to a substantial increase in the hardness of the formed component.

## Figures and Tables

**Figure 1 materials-16-04154-f001:**
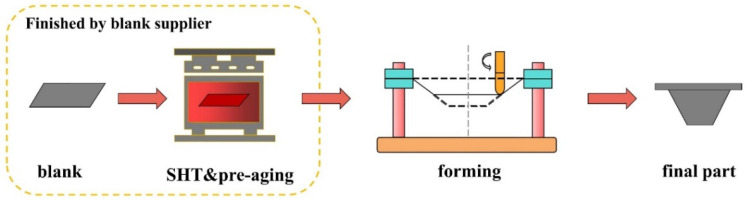
Diagram for the process chain of PH-SPIF.

**Figure 2 materials-16-04154-f002:**
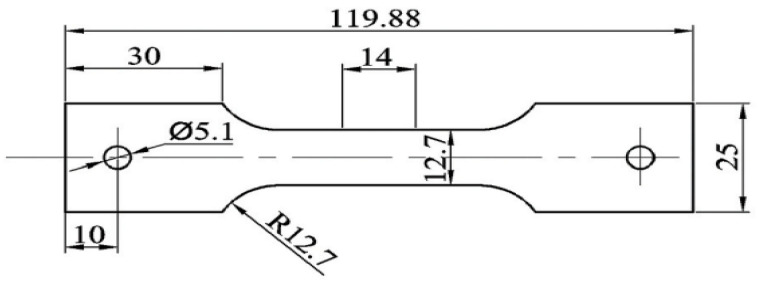
Specimen design for tensile tests (units: mm).

**Figure 3 materials-16-04154-f003:**
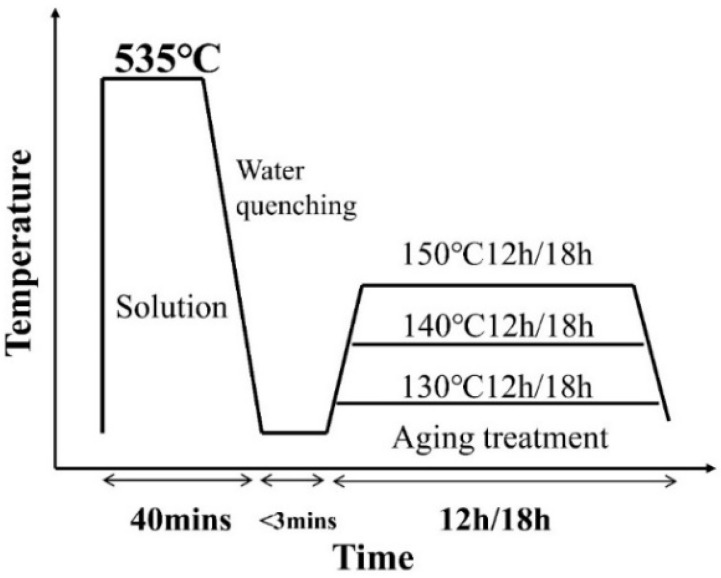
Processing flow of drawn parts sheet.

**Figure 4 materials-16-04154-f004:**
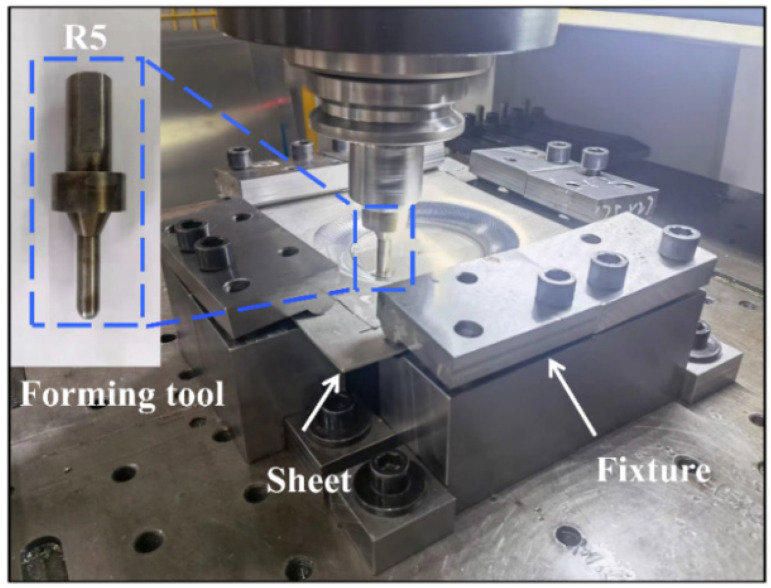
Experimental equipment.

**Figure 5 materials-16-04154-f005:**
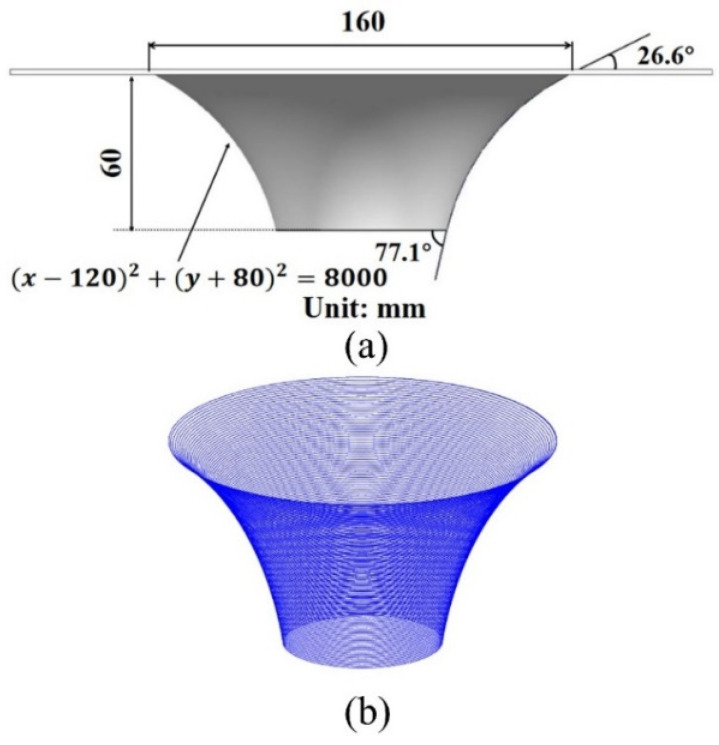
Schematic diagram of a variable angle conical forming part: (**a**) CAD model and geometrical details and (**b**) generated toolpath.

**Figure 6 materials-16-04154-f006:**
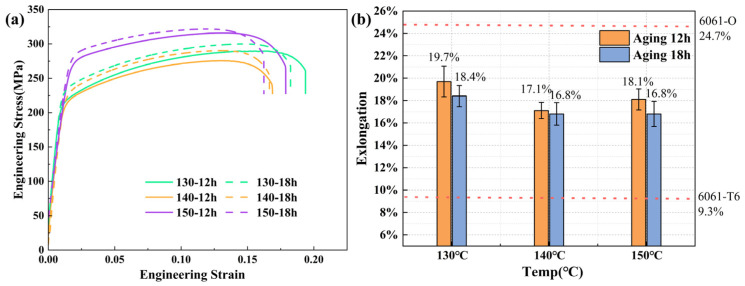
Tensile results of the plates with different parameters: (**a**) stress–strain curve and (**b**) elongation.

**Figure 7 materials-16-04154-f007:**
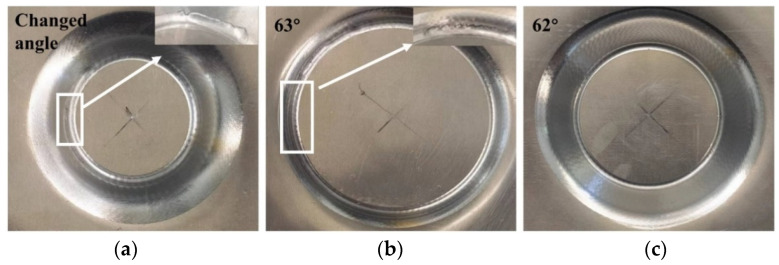
PA state formed parts with different wall angles: (**a**) Changed angle, (**b**) 63°, and (**c**) 62°.

**Figure 8 materials-16-04154-f008:**
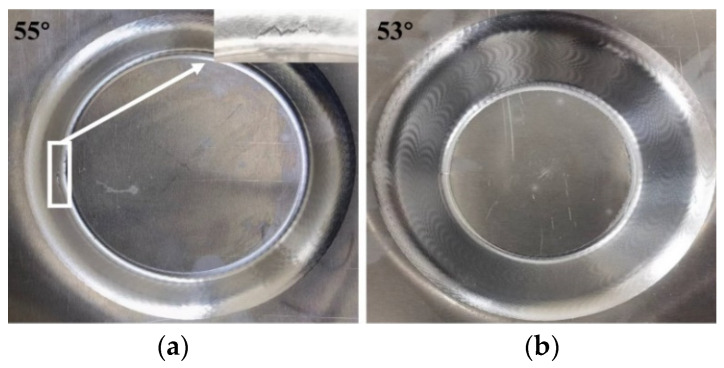
T6 state formed part: (**a**) 55°, (**b**) 53°.

**Figure 9 materials-16-04154-f009:**
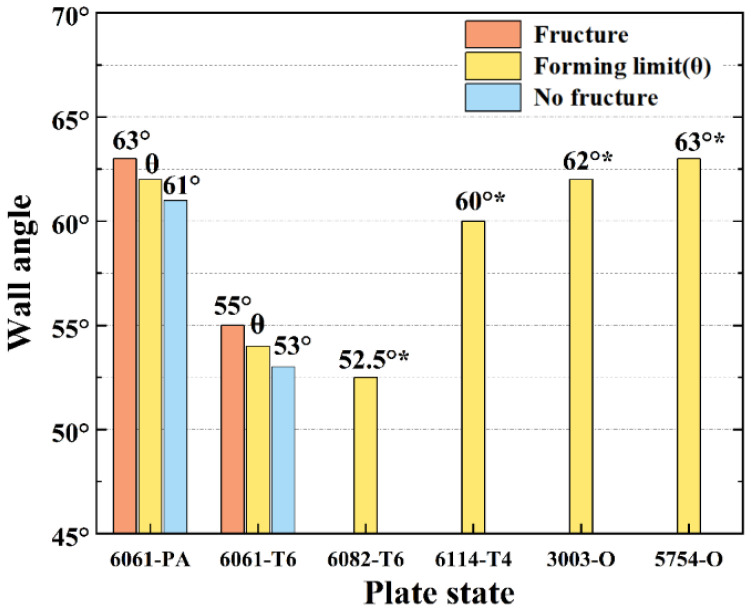
Forming results for sheets with different wall angles [28,29]. Data with * are sourced from literature.

**Figure 10 materials-16-04154-f010:**
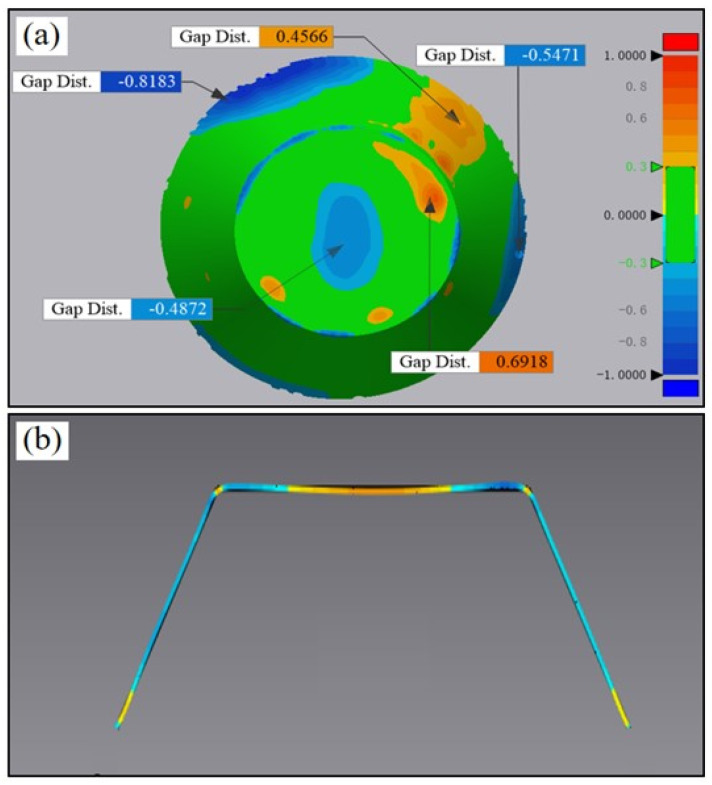
Comparison of molded part and CAD model: (**a**) 3D comparison and (**b**) 2D comparison.

**Figure 11 materials-16-04154-f011:**
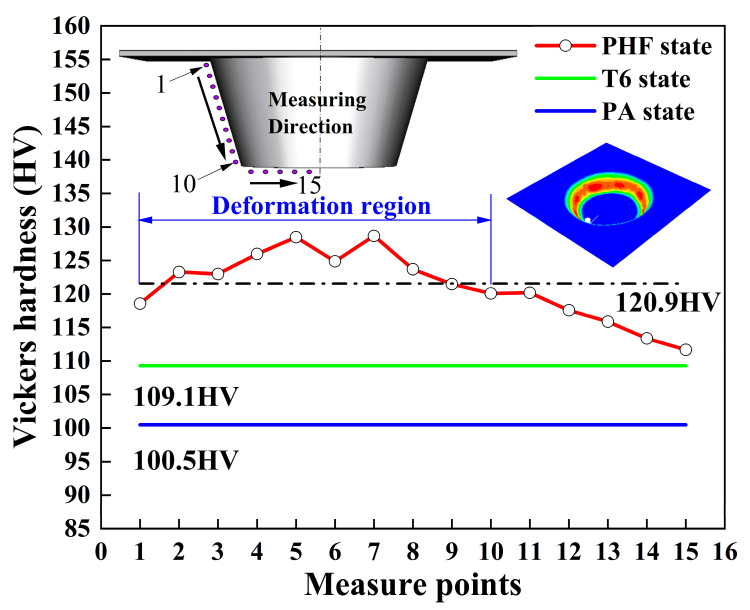
Hardness of T6, PA, PHF state sheet.

**Figure 12 materials-16-04154-f012:**
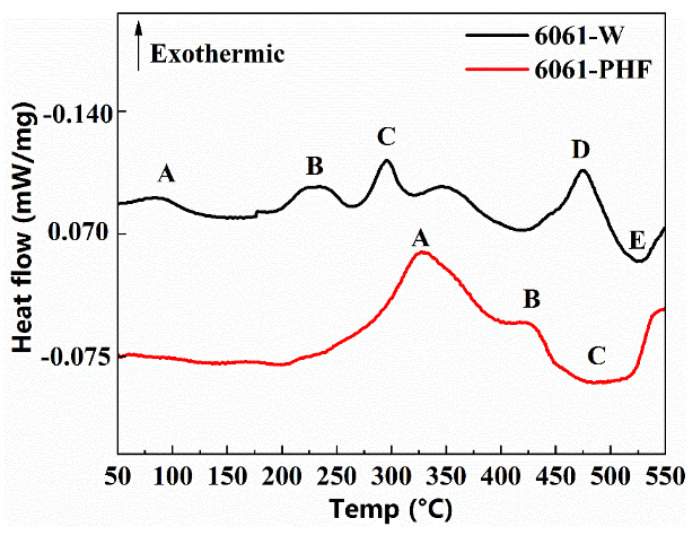
W-state and PHF-state DSC curves.

**Figure 13 materials-16-04154-f013:**
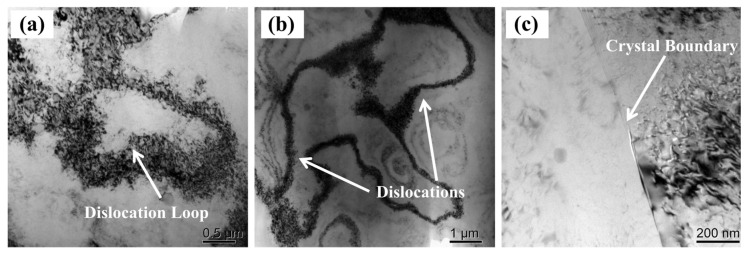
Dislocation distribution of formed components: (**a**,**b**) Dislocation loops; (**c**) Dislocations accumulate at grain boundary.

**Figure 14 materials-16-04154-f014:**
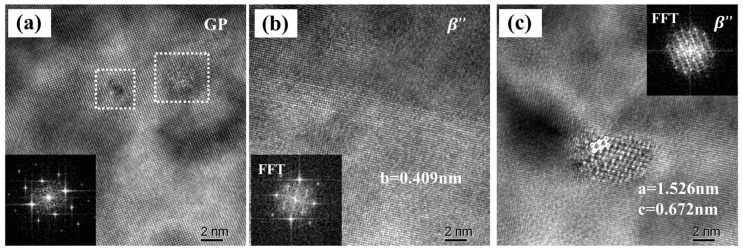
TEM observation of Al-Mg-Si alloy. The electron beam is parallel to [001] Al: (**a**) PA state HRTEM image with corresponding FFT (inset); (**b**,**c**) PHF state HRTEM image with corresponding FFT (inset).

**Figure 15 materials-16-04154-f015:**
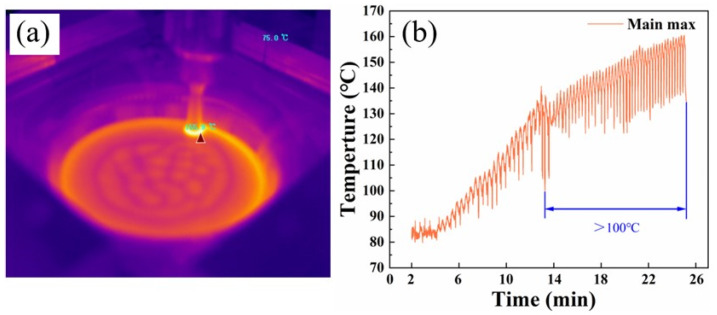
Forming temperature: (**a**) Thermographic image and (**b**) temperature–time curve.

**Table 1 materials-16-04154-t001:** Material chemical composition.

Composition	Si	Fe	Cu	Mn	Mg	Cr	Zn	Ti	Al
Content (wt%)	0.5	0.7	0.2	0.15	1.1	0.12	0.25	0.15	Bal

**Table 2 materials-16-04154-t002:** Material physical properties.

Density (Kg/ $m^{3}$)	Tensile Strength (MPa)	Yield Strength (MPa)	Elongation (%)	Young’s Modulus (GPa)	Poisson’s Ratio
2700	292	251	13.79	68.9	0.33

**Table 3 materials-16-04154-t003:** Main process parameters.

Parameter	AA6061-PA	AA6061-T6
Tool diameter [mm]	10	10
Tool feed rate [mm/min]	1500	1500
Tool vertical pitch [mm]	0.5	0.5
Tool rotational speed [rpm]	355	355
Conical frustum major diameter [mm]	160	160
Wall angle (θ)	55°–65°	50°–56°

## Data Availability

All data is available upon request.

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
