# Peer review of "Investigation of Pre-Aged Hardening Single-Point Incremental Forming Process and Mechanical Properties of AA6061 Aluminum Alloy"

_materials, 2023, doi:10.3390/ma16114154_

Round 1
Reviewer 1 Report
The paper proposes pre-aged hardening single point incremental forming process and mechanical properties of AA6061 aluminum alloy however, is necessary to clarify some process used to perform this analysis. Also, some factual clarifications will be helpful as highlighted in the comments below.
-What is the norm used to obtain the chemical composition?.
-What is the content of Al in the alloy?
-Poisson ratio of 3.3 is a typo mistake? If not please describe its physical meaning.
-What is the standard used for the tensile tests? What is the standard deviation?
-What are the equipment characteristics to reach the ramp described in figure 3?
-Please describe the use of table 3.What is it difference?
-How was defined the temperatures? What is the temperature reached in the raw material?
-The hardness measurements are the results of the ageing or the forming process?
-Improve the micrographies, the dislocations are not clear.
Please review the next papers and include it in your references.
-Ductile Fracture Behavior of Notched Aluminum Alloy Specimens under Complex Non-Proportional Load, Materials 2019, 12(10), 1598; https://doi.org/10.3390/ma12101598
- Design of an Aluminum Alloy Using a Neural Network-Based Model, Metals 2022, 12(10), 1587; https://doi.org/10.3390/met12101587
Reviewer 2 Report
Dear Autors.
In my opinion, the article raises quite interesting issues regarding pre-aged hardening of aluminium alloy an possibilities of point incremental forming process of this kind of material.
Unfortunately, before the material can be published, it needs major revisions:
11. The structure of the manuscript is not clear. For example, Discussions consists a lot of information, which should be found in sections Introduction or Methods (rows 243-279). In addition, the results of calorimetric and TEM studies do not fit into this section, as they rather show the processes occurring in the material, and do not confirm any theory in this case.
22. Information included in the sections Material and specimen design is not enough. It was not indicated what alloy it was in, in what condition it was delivered, in what condition the properties were given in Table 2. There is also no explanation for the further designation of the different states of the material which are used in the following. Markings PA, O, T6, PHF are not explained.
33. The device on which the hardness measurements were performed was not specified. How many measurements were made for each point?
44. Figure 9 contains the angles marked with a “*”, but the marking are not explained anywhere. It should be clarified.
55. The meaning of the results presented in Figure 10 is not clear. Was it a numerical model? Why is there no information about this earlier?
66. On Figure 11 numbering of points must be added in accordance with the numbering of measurement locations.
77. In section 4.2 a new previously unexplained sign W appeared.
88. Conclusions should contain information concerning only the article, but they contain too much unnecessary literature information.
99. A List of Abbreviations must be made.
Best regards.
Reviewer 3 Report
In this manuscript, AA6061 aluminum alloy is formed with different wall angles utilizing the pre-aged hardening single point incremental forming process (PH-SPIF). The investigation of forming limits, post-forming mechanical properties, and phase transformation during the forming process. The article's title is practical and attractive, but the following points should be considered before publishing.
The abstract should be written better and needs major revisions. The purpose of research and innovation should be clearly stated. Also, the performed tests should be presented first, and then the results should be presented quantitatively and qualitatively.
The article needs general writing and grammar editing. The number of keywords can be increased. The introduction is written very briefly, and at the end, a suitable summary of the importance of the present issue is not provided.
The introduction needs to be reformed and deepened. Use the following resources to complete this section. Effects of post-weld heat treatment on the microstructure and mechanical properties of laser-welded NiTi/304SS joint with Ni filler. Effect of heat input on interfacial characterization of the butter joint of hot-rolling CP-Ti/Q235 bimetallic sheets by Laser + CMT. Investigation of welding crack in micro laser welded NiTiNb shape memory alloy and Ti6Al4V alloy dissimilar metals joints.
What standard test is used to check the forming limit? Why is the forming limit diagram (FLD) not drawn after the incremental forming processes? Are the results presented in Table 2 extracted from the data sheet? For what purpose has the tensile test been repeated? In Figure 2: What standard test is provided? Tensile test conditions and other tests should be provided in more detail. How many tensile test samples were prepared for each group? How are the reproducibility of mechanical properties results checked? How accurate was the strain measurement?
Has a lubricant been used for the single point incremental forming process? The microhardness part is messed up (Lines 163-174). The scale bar and error bar is missing for some images and results. One of the influencing factors during incremental forming is the contact temperature, which is highly dependent on the pin speed. An increase in temperature can lead to an increase in forming limit. It is recommended to use higher speeds if possible.
The results section is well organized and categorized. But some parts of it are just reporting the results. Use the suggested resources to deepen the discussion. Investigation of mechanical properties, formability, and anisotropy of dual phase Mg–7Li–1Zn and The influence of post-annealing and ultrasonic vibration on the formability of multilayered Al5052/MgAZ31B composite. It is suggested to modify the conclusion section as well as the abstract.
No comment.
Round 2
Reviewer 1 Report
The requested changes have been done.
Reviewer 2 Report
The authors have answered the questions pointed out by the reviewer. The reviewer would like to suggest publication of this paper.